# Nano-in-Micro Delivery System Prepared by Co-Axial Air Flow for Oral Delivery of Conjugated Linoleic Acid

**DOI:** 10.3390/md17010015

**Published:** 2018-12-28

**Authors:** Qian Li, Fangfang Xue, Junle Qu, Liwei Liu, Rui Hu, Chenguang Liu

**Affiliations:** 1Key Laboratory of Optoelectronic Devices and Systems of Ministry of Education and Guangdong Province, College of Optoelectronic Engineering, Shenzhen University, Shenzhen 518060, China; liqian123@szu.edu.cn (Q.L.); jlqu@szu.edu.cn (J.Q.); liulw@szu.edu.cn (L.L.); 2College of Marine Life Sciences, Ocean University of China, Qingdao 266003, China; kikyoqq123@163.com

**Keywords:** co-axial air flow, nano-in-micro, gastrointestinal release

## Abstract

The preparation of a nano-in-micro delivery system (NiMDS) under mild conditions without using toxic organic solvents and expensive equipment still faces challenges. In this study, we introduced the co-axial air flow method to prepare NiMDS for the oral delivery of conjugated linoleic acid (CLA). The chitosan nanoparticles were prepared using the stearic-acid-modified chitosan through self-aggregation. Then, the chitosan nanoparticles were incorporated into alginate microparticles by the co-axial air flow method. The obtained chitosan nanoparticles and NiMDS were spherical in shape with the average sizes of 221–243 nm and 130–160 μm, respectively. Compared with alginate microparticles, the hybrid particles were of fewer fragments, were bigger in size, had a higher mechanical strength, and showed a controlled release in the phosphate buffer solution (pH 1.2 or 7.4). The release kinetics study showed that encapsulating the chitosan nanoparticles into the alginate microparticles inhibited the dissolution of alginate microparticles at the initial stage. These results revealed the potential of NiMDS as an ideal oral carrier for the sustained release of CLA in the gastrointestinal environment.

## 1. Introduction

Micro- or nano-sized particles have found numerous applications as oral delivery vehicles for lipid nutraceuticals. An ideal delivery vehicle for the lipid nutraceuticals should have the following advantages such as the sustained release in the gastrointestinal tract, increasing water dispersion, bioavailability of lipid nutraceuticals, and good compatibility of delivery vehicles [1,2]. The gastrointestinal tract has complicated environments including a strong acidic stomach and a weak alkaline intestine. Sophisticated delivery systems that incorporate the advantages of both nanoparticles and microparticles have been introduced to replace simple nanoparticles or microparticles alone. Nano-in-micro hybrid particles contain nanoparticles encapsulated within the microparticles, and thus combine two different properties in one system, which make them a good candidate for the gastrointestinal sustained release [3,4]. Several methods have been developed for the preparation of hybrid particles, such as emulsion polymerization, intercalative polymerization, hybrid latex polymerization, spray-drying, and supercritical fluids technology [5,6,7]. However, the first three use surfactants and organic solvents, which are deterrent in toxicity. Spray-drying utilizes high temperature while supercritical fluids technology can be quite expensive. Hence, co-axial air flow can be considered as the most preferred technique owing to its capability of forming small microsphere size, possibility to scale up and reduction in surfactants and oils [8]. Several reports have been focused on preparing microparticles with the co-axial air flow method; however, until now, only very few reports on preparing nano-in-micro delivery vehicles are available.

The pH-sensitive hydrogels have attracted much attention as oral delivery vehicles for the protection against gastric acid and promotion of the intestinal absorption. Alginate gels are particularly suitable for oral applications due to their pH-sensitive properties. Shrinking of the alginate gels at low pH (gastric conditions) and their dissolution under intestinal conditions would avoid release in the stomach and allow specific release in the intestine [1]. Chitosan, which swells in stomach and shrinks in intestine, is complementary to alginate. Hybrid alginate–chitosan matrix is a good candidate for the controlled release of the lipid nutraceuticals in the gastrointestinal environment [9,10]. It is of great value to prepare the alginate–chitosan hybrid matrix with a new method under mild conditions without toxic solvents.

Conjugated linoleic acid (CLA), a family of the essential fatty acids for humans, is of great importance in health and disease prevention [11]. It induces diverse biological activities in humans and potential health benefits in experimental animals, including anti-inflammatory, anti-cancer, and anti-atherosclerotic activities, and, importantly, reducing body fat content [12,13,14,15]. As the lining of the digestive tract is aqueous in nature, to be assimilated, it is important to enhance the water dispersion of lipid nutraceuticals. In the present study, water-soluble chitosan nanoparticles with a fatty core were used to increase the water dispersion of CLA, and CLA-loaded chitosan nanoparticles were encapsulated into the alginate microparticles to achieve a controlled release in the stomach and the intestine.

## 2. Results and Discussion

### 2.1. Characterization of Amphiphilic Chitosan

In order to obtain amphiphilic chitosan, hydrophobic stearic acid was conjugated to the hydrophilic chitosan via amide bond under the activation by 1-(3-dimethylaminopropyl)-3-ethyl carbodiimide (EDC) and *N*-Hydroxysuccinimide (NHS) (Figure 1a). The proton nuclear magnetic resonance (^1^HNMR) spectra in Figure 2 verified the conjugation of stearic acid to the chitosan. The basic components of chitosan are shown in the peaks of 3.0 ppm (H_2_), 3.4–3.8 ppm (H_3_, H_4_, H_6_, H_6,_), 4.4 ppm (H_5_), and 4.7 ppm (H_1_). The new peaks in chitosan–stearic acid conjugate showed the presence of stearic acid groups at 1.2 ppm (CH_2_) and 2.7 ppm (chemical shift of CH_2_, which is close to C=O) [1,16]. The degree of substitution, which was defined as the number of stearic acid groups per hundred sugar residues, was 13.34, 18.02, or 21.36 determined by the elemental analysis. The chitosan–stearic acid was named CS–SA13, CS–SA18 or CS–SA21, respectively.

The amphiphilic polymers begin to form the hydrophobic domains in the aqueous media through intermolecular hydrophobic interactions at a certain concentration, which was defined as the critical aggregation concentration. As shown in Table 1, the critical aggregation concentration values of amphiphilic chitosan are 0.037, 0.031, and 0.027 mg/mL, respectively. With the increasing degree of substitution in the conjugates, the critical assembly concentration (CAC) values became progressively smaller, which was attributed to the increase in hydrophobicity [17].

### 2.2. Characterization of Chitosan Nanoparticles

The critical aggregation concentration is defined as the threshold concentration of self-aggregation of amphiphilic polymers by intra- and/or intermolecular hydrophobic interactions. As shown in Table 1, critical aggregation concentration decreased from 0.037 mg/mL to 0.027 mg/mL with the increasing of degree of substitution. This indicated that higher degree of substitution enhanced the hydrophobic interactions and facilitated the formation of nanoparticles. The TEM image of CLA-loaded chitosan nanoparticles shown in Figure 2a indicates that the prepared nanoparticles arespherical in shape and have a good structural integrity with a size of about 200 nm. The mean hydrodynamic diameters of the particles were 265.7 ± 7.5, 238.8 ± 5.8, and 219.0 ± 6.3 nm (Table 1), which matched with the result of TEM. The size of self-aggregates decreased as the degree of substitution increased, indicating the formation of denser hydrophobic cores in the higherdegree of substitution sample. The reason may be that the stronger intra- and intermolecular hydrophobic interactions between hydrophobic stearic acid grafts lead to denser aggregates. All nanoparticles possessed positive surface charges of 21–28 mV.

### 2.3. Characterization of Nano-in-Micro Delivery System (NiMDS)

The preparation method of NiMDS is briefly illustrated in Figure 1b. After chitosan polymers self-assembled into nanoparticles under the ultrasound, chitosan nanoparticles and alginate sodium were mixed and subjected to co-axial air flow method to form NiMDS. To stabilize the NiMDS, alginate were cross linked by calcium ion after air spray. The successful incorporation of nanoparticles into microparticles was confirmed by FTIR and laser confocal microscope. FTIR in Figure 2j showed chitosan nanoparticles representing a peak at 1649 cm^−1^ (C=O stretching vibration of amide bond of chitosan) and alginate microparticles representing peak at 1619 cm^−1^ (asymmetric stretch of carboxyl groups of alginate) [1,18,19]. The increased peak at 1639 cm^−1^ with a slight shift in the nano-in-micro particles indicated lapping peaks of amide bonds between the chitosan particles and the carboxyl groups of alginate. In addition, the peak at 3445 cm^−1^ (–OH stretching vibration) in alginate microparticles shifted to 3401 cm^−1^ in the NiMDS, which likely indicated the presence of hydrogen bonds between the chitosan nanoparticles and the alginate microparticles [20]. Hence, we speculated that the process by which the chitosan nanoparticles were encapsulated into the alginate matrix included not only simple physical encapsulation but also hydrogen bond interactions. Figure 2b showed that green fluorescence of FITC–chitosan nanoparticles uniformly dispersed in alginate microparticles, indicating successful encapsulation of nanoparticles into microparticles. This could facilitate regular release of encapsulated molecules [21].

The size and morphology of NiMDS is shown in Figure 2 and Table 2. The hybrid particles are rounder and have fewer fragments than alginate microparticles (Figure 2), which can be attributed to an increased viscosity from 89.70 ± 0.65 to 104.71 ± 0.46 after adding the nanoparticles into alginate solutions (Table 2). As the surface-tension-driven contraction of the deformed droplets, which leads to the formation of spherical-shaped beads, becomes slower with the increasing alginate solution viscosity, the time of falling of the droplets into the gelation bath is no more sufficient to reshape the deformed droplets to a sphere [8]. As the concentration of chitosan nanoparticles increased, the size of the hybrid particles increased from 130.21 μm to 153.26 μm (Table 2). After lyophilization, the SEM showed that NiMDS were stiffer than microparticles, indicating nanoparticles acted as skeleton in microparticles.

### 2.4. In Vitro Release Studies of Loaded CLA

Loading efficiency and loading capacity are shown in Table 2. With the increasing concentration of chitosan nanoparticles, the loading efficiency increased from 84.05% to 95.65%, and the loading capacity of the hybrid particles increased from 0.17% to 0.31%. This result indicated that more nanoparticles in NiMDS provide more molecule loading sites. Figure 3a shows the in vitro release profiles of CLA from chitosan nanoparticles and the NiMDS in simulated gastrointestinal conditions at 37 °C for 10 h. The samples were first placed in a buffer (pH 1.2) for 3 h and then in intestinal fluid buffer (pH 7.4) for 7 h. A rapid burst of CLA from the chitosan nanoparticles was observed in the first 3 h during which approximately 80% of the CLA was released. The release rate then gradually decreased, and a final cumulative release of nearly 97% was achieved by 10 h. In comparison, the release rate was much lower for the hybrid particles. Only approximately 30–40% CLA was released from within the first 3h, while approximately 60–70% CLA was released by 10 h.

### 2.5. Release Kinetics

In order to explainthe mechanism of reduced burst and controlled release of CLA from the NiMDS, release kinetics in pH7.4 was used as a model. Based on the CLA release results, the release mechanisms were estimated and discussed through various common models of drug release from porous polymers, such as zero-order kinetic model, first-order kinetic model, the Higuchi model, and the Regter–Pappersis model. Figure 3b showed that the release of CLA from alginate microspheres was observedin the initial 2 h, whereas CLA release from the NiMDS was observed throughout the entire process. Hence, to reveal the mechanism of drug release, the drug release profiles were divided into two release phases, 0–2 h and 2–7 h. The data of the two release phases were fitted according to the kinetic models (Table 3). It was found that CLA release profiles of S0 in 0–2h, S3 in 0–2h and S3 in 2–7h can be better agreed with the first-order, Ritger–Peppas and Higuchi equations, respectively. To investigate the release mechanisms of CLA, Ritger–Peppas equation was utilized to compare the three samples and they all showed an R2 > 0.99. In the case of S0 in 0–2h and S3 in 2–7h, the Fickian diffusion process *n* was 0.7406 and 0.7338 (0.43 < *n* < 0.85), respectively, indicating that the release was a combination of Fickian diffusion and dissolution [22]. In the case of S3 in 0–2h, the Fickian diffusion process *n* was 0.4168 (*n* < 0.43), which suggested the diffusion release mechanism. Therefore, encapsulating the chitosan nanoparticles into alginate microparticles could effectively inhibit the dissolution of alginate microparticles at the initial stage.

### 2.6. In vitro Cell Viability Assay

To investigate the cytotoxicity, chitosan nanoparticles, alginate microparticles (S0), and the NiMDS (S1–S3) were incubated with Caco-2 cells for 24 h, and the cell viability was measured via 3-(4,5-dimethylthiazol-2-yl)-2,5-diphenyltetrazolium bromide (MTT) assays. Figure 4 shows that the cell viability was above 95% for all groups when the concentrations of nano-, micro-, nano-in-micro particles ranged from 50 to 1000 μg/mL, and there was no difference in cell viability between chitosan nanoparticles, alginate microparticles and the NiMDS (*p* > 0.05). These results suggest that the nanoparticles did not cause significant cytotoxicity to cells under normal concentrations. Thus, the NiMDS induced no cytotoxicity and holds a great potential for use as a new drug carrier.

## 3. Materials and Methods

### 3.1. Materials

1-(3-dimethylaminopropyl)-3-ethyl carbodiimide (EDC), *N*-Hydroxysuccinimide (NHS) 3-(4,5-dimethylthiazol-2-yl)-2,5-diphenyltetrazolium bromide (MTT), stearic acid, CLA, alginate and chitosan were obtained from Sigma Chemicals. Stearic acid was supplied by Bodi Chemical Engineering Co., Ltd. (Tianjin, China). All other chemicals used in this study were of analytical grade.

### 3.2. Synthesis and Characterization of Amphiphilic Chitosan

Amphiphilic chitosan was prepared by conjugating stearic acid to chitosan via an amide bond. Chitosan was first dissolved in 1 wt% acetic acid, and different amounts of stearic acid were mixed with chitosan according to the predetermined molar ratios of 1:7, 1:3, and 1:2. Then, EDC (ten times the molar number of stearic acid) was dissolved in 20 mL ethanol solution and gradually added to the mixture solution. After stirring for 5 h at 400 rpm and 80 °C, the solution was left at room temperature, followed by stirring for another 24 h. Finally, the reaction solution was dialyzed against distilled water and then lyophilized. The lyophilized product was further purified with ethanol to remove the byproduct. Finally, the chitosan–stearic acid product was re-dispersed in distilled water and lyophilized, and the degree of substitution of chitosan–stearic acid was determined by elemental analysis. Fourier transform infrared spectroscopy (FTIR) spectra were recorded in the range of 400–4000 cm^−1^ with a resolution of 4 cm^−1^ as KBr pellets. ^1^H NMR spectra of the samples were recorded using a Bruker ARX 300 spectrometer at 25 °C. The sample was dissolved in 1% C_2_D_6_OS of D_2_O solution (*v*/*v*) to yield a concentration of 2 mg/mL. The measurement conditions were as follows: a spectral window of 500 Hz, 32 k data points, a pulse angle of 30, an acquisition time of 2.03 s, and 32 scans with a delay of 1 s between each scan.

### 3.3. Preparation of CLA-Loaded Nanoparticles

CLA-loaded chitosan–stearic acid nanoparticles were self-assembled by ultrasound. Briefly, 10 mg of chitosan–stearic acid was dissolved in 10 mL 1% acetic acid aqueous solution, followed by the addition of CLA–alcohol solution (1 mg/mL). The mixture was subsequently treated with a 2s on/3s off ultrasound cycle for 2 min in an ice bath using a probe-type sonifier, which was repeated three times.

The critical aggregation concentration of the synthesized chitosan–stearic acid was estimated by fluorescence spectroscopy using pyrene as a probe. A fluorometer (F-2500, HITACHI Co., Tokyo, Japan) was used to record the fluorescence spectra with an excitation wavelength of 337 nm, and the slits were set at 2.5 nm (excitation) and 10 nm (emission). The intensities of the emission were monitored over the wavelength range of 360–450 nm. The concentration of chitosan–stearic acid micelle solutions containing 5.93 × 10^−7^ M pyrene was varied from 5.0 ×10^−3^ to 1.0 mg/mL. Then, the ratio of the intensities (I_1_/I_3_) of the first peak (I_1_, 374 nm) to the third peak (I_3_, 385 nm) was calculated.

### 3.4. Preparation of CLA-Loaded NiMDS

The NiMDS was prepared using an air-driven droplet generator (Nisco Engineering Inc., Zurich, Switzerland), as shown in Figure 1. The process involved mixing the CLA-loaded nanoparticle (NP) suspension or CLA in a 1.5% (*w*/*v*) sodium alginate solution. The NP–alginate suspension was then dispersed into uniform droplets sprayed through the nozzle after optimizing several instrumental parameters (nozzle diameter, flow rate, and air pressure) and sample parameters (concentration of alginate, concentration of CaCl_2_, and the ratio of nanoparticles). The flow rate and air pressure were monitored and fixed according to the in-built program of the syringe pump. Several optimization steps were carried out to set the instrumental parameters, including the nozzle diameter (250 μm), flow rate (90 mL/h), and pressure (18,000 bar). The fine droplets of alginate solution/suspension were collected into a 0.2 M CaCl_2_ solution for gelation under constant stirring (200 rpm) for 1 h. The detailed process is not disclosed in this paper. The loaded microspheres obtained were separated by centrifugation and washed using double distilled water.

### 3.5. Characterization of Nanoparticles and NiMDS

The sizes and zeta potential of the chitosan nanoparticles and the NiMDS were measured using a 3000HSZetasizer (Malvern Instruments, Worcestershire, UK) at a detector angle of 90° and at 633 nm. The surface morphology of chitosan nanoparticles was inspected by transmission electron microscopy (TEM) (JEM-2010, Tokyo, Japan). The surface morphology of the NiMDS was investigated by scanning electron microscopy (SEM). The viscosity of the mixture of chitosan nanoparticles and alginate solution was determined using anUbbelohde viscometer at 25 °C.

To monitor the distribution of chitosan nanoparticles in microparticles, chitosan–stearic acid conjugates were labeled with FITC. Fifty milligrams chitosan–stearic acid conjugate and 50 mg FITC were dissolved in 10 mL heavy carbon buffer (0.1 mol/L, pH 9.5) for 24 h at room temperature followed by dialysis (MWCO 8000–14,000 Da) against excess deionized water for 5 days at 4 °C. The FITC–chitosan nanoparticles were finally obtained by freeze-drying and were enclosed by alginate microparticles using an co-axial air flow (Nisco Engineering Inc., Zurich, Switzerland), as described in Section 2.4. Fluorescence image of the NiMDS were obtained with confocal laser scanning microscope (CLSM) equipped with a krypton–argon laser (Olympus FluoViewTM, Tokyo, Japan). The standard filter settings for fluorescence excitation (488 nm) and emission (520 nm) were used.

### 3.6. CLA Loading Efficiency and Loading Capacity

The non-encapsulated CLA was separated by centrifugation and measured with spectrophotometry (Shimadzu UV–Visible Spectrophotometer UV-1603, Tokyo, Japan) at 234 nm. The loading efficiency and loading capacity were calculated as follows (1) and (2):Loading efficiency% = (W_1_ − W_0_)/W_1_ × 100(1)
Loading capacity% = (W_1_ − W_0_)/polymer weight × 100(2)
where W_1_ is the total amount of CLA, and W_0_ is the free CLA in the supernatant.

### 3.7. In Vitro Release Study

In vitro release studies of the alginate microspheres, chitosan nanoparticles, and CLA-loaded NiMDS were performed as follows—hydrochloric acid of pH 1.2 for 3 h and then buffer solution of pH 7.4 for 7 h. A known quantity of lyophilized particles was briefly dispersed in 50 mL of the dissolution medium under magnetic stirring (60 rpm, 37 °C). Then, 1 mL of the sample was periodically withdrawn and centrifuged for 20 min at 15,000 rpm at 4 °C. The released CLA in the supernatant was measured using a UV–Visible spectrophotometer at 234 nm.

### 3.8. Release Kinetics

To study the CLA release mechanism from the alginate microparticles (S0) and the NiMDS (S1), four models were considered to fit the experimental release data during two different time periods (0–2 h and 2–7 h), and the models are listed below.

Model 1 was proposed based on zero-order regression (3),
*Q* = *Q*_0_ + *Kt*(3)
where *Q*_0_ is the initial release quantity, *Q* is the amount released, *K* is the zero-order release rate constant, and *t* is the release time.

Model 2 was proposed based on first-order regression (4),
*Q* = *Q*_∞_[1 − (Q/Q_∞_) × e^−*Kt*^](4)
where *K* is the first-order release rate constant, $Q/Q_{\infty }$ is the fractional drug release, and *e* is an empirical value.

Model 3 corresponding to Higuchi’s plot regression can be expressed as follows (5):*Q/Q*_∞_ = *Kt*^1/2^(5)

Model 4 is described by the well-known Ritger–Peppas equation, which is more comprehensive (6):*Q/Q*_∞_ = *Kt*^n^(6)
where *Q/Q*_∞_ is the fractional drug release, *K* is a constant indicating structural and geometric characteristics, t is the release time, and n is the release exponent that is related to the drug release mechanism. All data were analyzed using Origin Pro 8 (Origin Lab Corporation, Northampton, MA, USA).

### 3.9. In Vitro Cytotoxicity Assay

The potential toxic effects of the chitosan nanoparticles, alginate microparticles, and NiMDS were evaluated with the MTT method. Caco-2 cells were seeded on 96-well plates at a density of 1 × 10^4^ cells per well and were cultured for 24 h. MTT solution was added to mixtures containing different concentrations of chitosan nanoparticles, alginate microparticles, or the NiMDS and cultivated for 24 h. The mixtures were then supplemented with 100 μL DMSO, and the absorbance was measured at 490 nm.

## 4. Conclusions

Co-axial air flow is the ideal method to prepare nano-in-micro particles for encapsulating hydrophobic CLA, and chitosan in alginate matrix is suitable for the sustained release of CLA in the gastrointestinal environment. The chitosan nanoparticles embedded in alginate microparticle matrix showed fewerfragments, bigger size, higher mechanical strength, lower burst release, and a controlledrelease in the phosphate buffer solution (pH 1.2 or 7.4) than chitosan nanoparticles or alginate microparticles alone.

## Figures and Tables

**Figure 1 marinedrugs-17-00015-f001:**
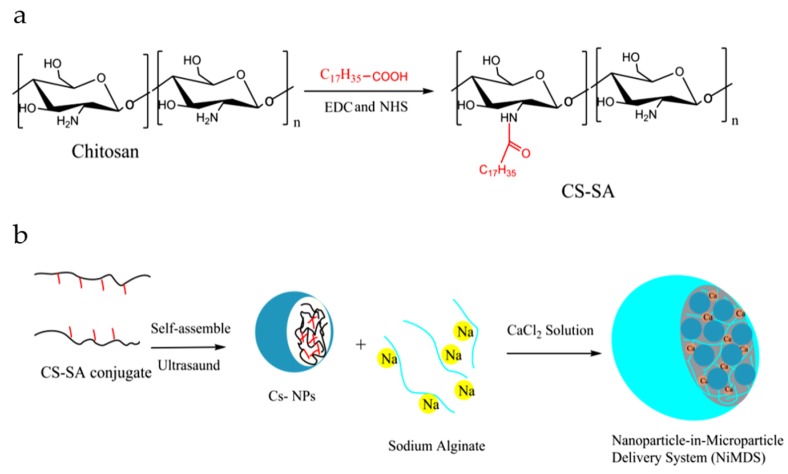
Schematic illustrations of the (**a**) modification of chitosan with stearic acid; and (**b**) formation of nano-in-micro delivery system (NiMDS).

**Figure 2 marinedrugs-17-00015-f002:**
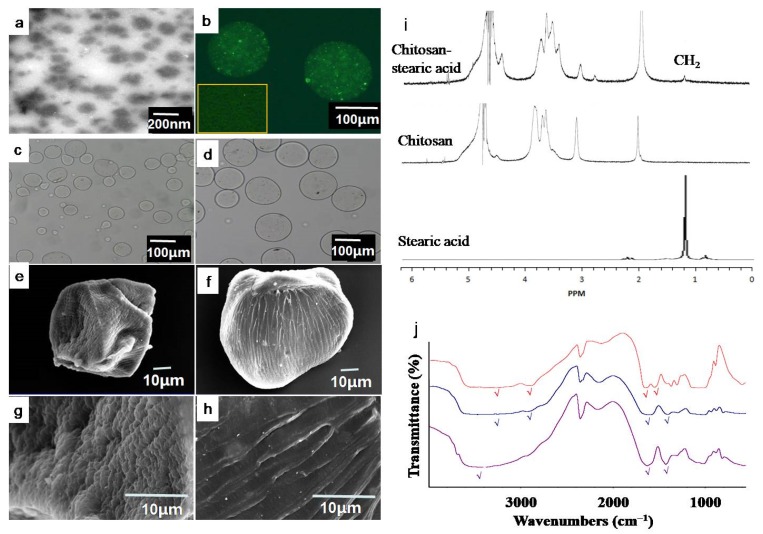
Characterization of amphiphilic chitosan, chitosan nanoparticles, alginate microparticles, and NiMDS. (**a**) TEM image of chitosan nanoparticles; (**b**)fluorescence micrograph of the fluorescein isothiocyanate (FITC) labeled chitosan nanoparticles (inserted picture) and FITC labeled chitosan nanoparticles in alginate microparticles; (**c**) micrographs of alginate microparticles; (**d**) micrographs of NiMDS; (**e**) SEM images of panorama of alginate microparticles; (**f**) SEM images of panorama of NiMDS; (**g**) surface morphology of alginate microparticles; (**h**) surface morphology of NiMDS; (**i**) NMR of amphiphilic chitosan, chitosan, and stearic acid; and (**j**) FTIR of chitosan nanoparticles (red) and alginate microparticles (blue) and NiMDS (purple).

**Figure 3 marinedrugs-17-00015-f003:**
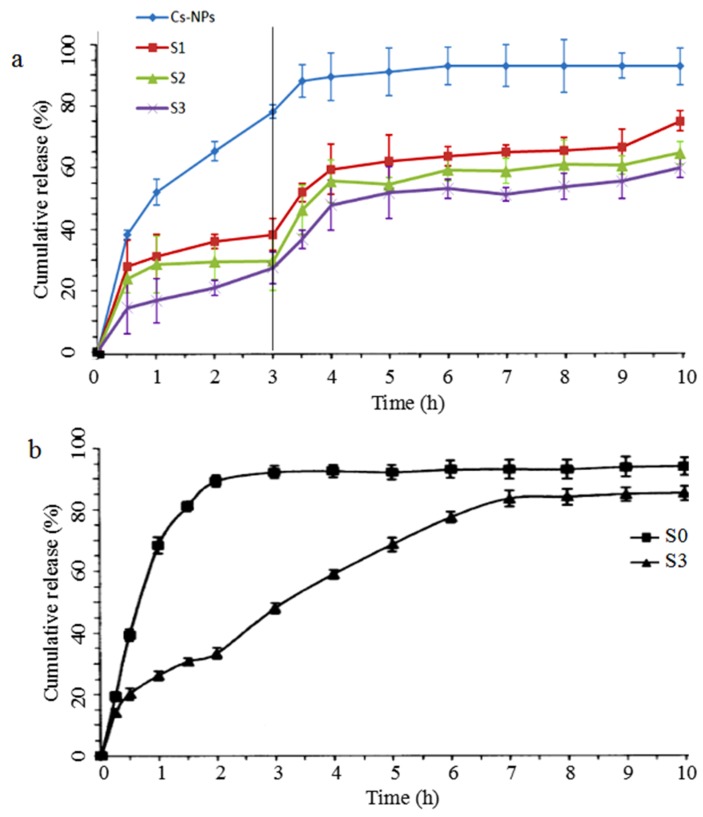
In vitro release of CLA from nanoparticles and the NiMDS. (**a**) Release of CLA in pH1.2 and pH7.4 phosphate buffer subsequently; and (**b**) release of CLA in pH7.4 phosphate buffer.

**Figure 4 marinedrugs-17-00015-f004:**
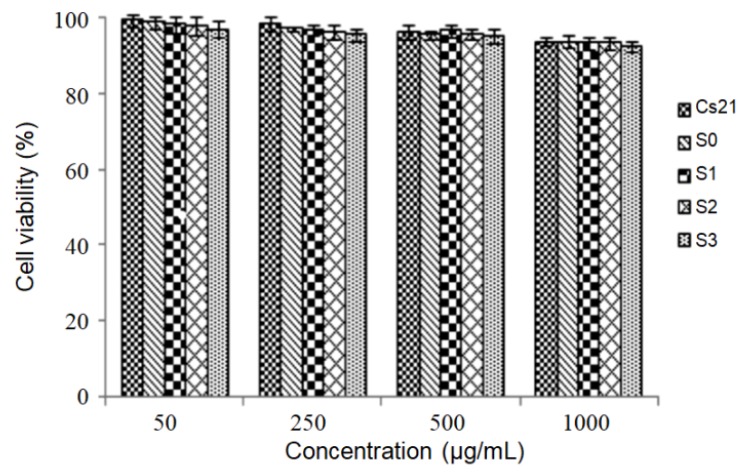
Cell viability after 24 h incubation with chitosan nanoparticles (CS21), alginate microparticles (S0), and the NiMDS (S1, S2, S3) with the concentration of nano-, micro-, nano-in-micro particlesvarying from 50 to 1000 μg/mL.

**Table 1 marinedrugs-17-00015-t001:** Characterization of chitosan nanoparticles.

Sample	Degree of Substitution (%)	Critical Aggregation Concentration (mg/mL)	Size (nm)	PDI	Zeta Potential (mV)
CS–SA13	13.34	0.037	265.7 ± 7.5	0.2 ± 0.1	27.8 ± 0.8
CS–SA18	18.02	0.031	238.8 ± 5.8	0.3 ± 0.1	25.5 ± 1.7
CS–SA21	21.36	0.027	219.0 ± 6.3	0.3 ± 0.1	23.6 ± 0.1

**Table 2 marinedrugs-17-00015-t002:** Characterization of the conjugated linoleic acid-loaded NiMDS.

Sample	Chitosan Nanoparticles:Alginate (mL:mL)	Size (μm)	Relative Viscosity (η)	Loading Efficiency (%)	Loading Capacity (%)
S0	0:10	70.99 ± 2.575	89.70 ± 0.65	80.12 ± 0.95	0.15 ± 0.02
S1	2.0:10	130.21 ± 6.209	99.13 ± 0.33	84.05 ± 0.65	0.17 ± 0.02
S2	2.5:10	133.90 ± 5.304	102.41 ± 0.74	88.96 ± 1.04	0.22 ± 0.01
S3	3.0:10	153.26 ± 9.324	104.71 ± 0.46	95.62 ± 1.20	0.31 ± 0.01

**Table 3 marinedrugs-17-00015-t003:** Kinetics values obtained from alginate microparticles-S0 and NiMDS-S3.

Sample	Zero-Order R^2^	First-Order R^2^	Higuchi’s Plot R^2^	Ritger-Peppas	
				R^2^	*n* Value
S0 0-2h	0.9288	0.9985	0.9734	0.9695	0.7406
S3 0-2h	0.8403	0.8816	0.9865	0.9903	0.4168
S3 2-7h	0.9816	0.9951	0.9975	0.9922	0.7338

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
