# Peer review of "Nano-in-Micro Delivery System Prepared by Co-Axial Air Flow for Oral Delivery of Conjugated Linoleic Acid"

_marinedrugs, 2018, doi:10.3390/md17010015_

Round 1
Reviewer 1 Report
The manuscript "Nano-in-micro delivery system prepared by co-axial 2 air flow for oral delivery of conjugated linoleic acid" is an interesting work that investigates the preparation of nano-in-micro delivery systems using a co-axial air flow method. The nano-in-micro delivery systems encapsulate nanoparticles within microparticles and were designed for oral delivery of conjugated linoleic acid. Chitosan nanoparticles with hydrophobic cores were prepared through self-aggregation of stearic acid modified chitosan. The obtained nanoparticles were encapsulated into alginate microparticles to realize controlled released in stomach and intestine. The topic of the research is interesting and the results are exhaustively presented but some information need to be added. I recommend minor revision before publication in Marinedrugs.
In the abstract, lines 13-16, the sentences are unclear and this makes difficult to the readers to understand the actual composition and preparation of nano-in-micro delivery systems.
In the Introduction, lines 28-29, the sentence “increasing water solubility of lipid nutraceuticals” is not correct because vehicle increase the bioavailability of lipid nutraceuticals not the solubility.
In the Introduction, line 48, the word “intestinal” should be re-place with “intestine”
In the Introduction, lines 55-58, the sentences “it is important to enhance water solubility of lipid nutraceuticals. In the present study, water-soluble chitosan nanoparticles with a fatty core were used to increase water solubility of CLA” are not correct because the loading into nano- or micro-particles does not affect the CLA solubility but provide its water dispersion and bioavailability.
In the result and discussion, several abbreviations are reported which make difficult the understanding of the text. E.g lines 82-83, CAC and DD should be reported as whole words.
In the result and discussion, line 100, the word “certificated” should be re-place with “confirmed”
In the result and discussion, line 128, the word “drug” should be re-place with “loaded molecule”
In the result and discussion, lines 165-66, please report in the figure 4 and legend “concentration of ?” it is the concentration of CLA?
Author Response
We appreciate the valuable advices that have helped us to improve the manuscript. Changes to the manuscript have been listed as follows.
1. In the abstract, lines 13-16, the sentences are unclear and this makes difficult to the readers to understand the actual composition and preparation of nano-in-micro delivery systems.
The sentences in lines 13-16 have been changed to “Chitosan nanoparticles were prepared using stearic acid modified chitosan through self-aggregation. Then chitosan nanoparticles were incorporated into alginate microparticles with co-axial air flow method. ”
2. In the Introduction, lines 28-29, the sentence “increasing water solubility of lipid nutraceuticals” is not correct because vehicle increase the bioavailability of lipid nutraceuticals not the solubility.
“increasing water solubility of lipid nutraceuticals” has been changed to “increasing water dispersion and bioavailability of lipid nutraceuticals” in lines 28-29.
3. In the Introduction, line 48, the word “intestinal” should be re-place with “intestine”
“intestinal” has be re-place with“intestine”in line 48.
4. In the Introduction, lines 55-58, the sentences “it is important to enhance water dispersion and bioavailability water solubility of lipid nutraceuticals. In the present study, water-soluble chitosan nanoparticles with a fatty core were used to increase water solubility of CLA” are not correct because the loading into nano- or micro-particles does not affect the CLA solubility but provide its water dispersion and bioavailability.
“increase water solubility” has been changed to “increase water dispersion” in lines 55-58.
5. In the result and discussion, several abbreviations are reported which make difficult the understanding of the text. E.g lines 82-83, CAC and DD should be reported as whole words.
“CAC” has been changed to “critical aggregation concentration”
“DS” has been changed to “degree of substitution”
“LC” has been changed to “loading capacity”
“LE” has been changed to “loading efficiency”
6. In the result and discussion, line 100, the word “certificated” should be re-place with “confirmed”
certificated” has be replaced with “confirmed” in line 100.
7. In the result and discussion, line 128, the word “drug” should be re-place with “loaded molecule”
“drug” has been replaced with “loaded molecule” and “drug” has been replaced with “CLA” in line 148
8. In the result and discussion, lines 165-66, please report in the figure 4 and legend “concentration of ?” it is the concentration of CLA?
“concentration of nano-, micro-, nano-in-micro particles” has been added in lines 165-166.

Reviewer 2 Report
The authors are expert on to use the nano-in-micro delivery system prepared by co-axial air flow. They have now applied the method for oral delivery of CLA that can be considered a marine drug.
The study is well done and the results interesting.
General comment: English should be revised over all the manuscript
For example, line 26 change has for have
Line 45 to thier properties
Line 53 including exerting
Line 140. In order to explanation
Line 274 to prepared
Introduction I would avoid the term in gastrointestinal fluid since the study was done in vitro just changing the pH of the medium. Gastrointestinal fluid has a complex composition that cannot be compared with the vehicle used in these experiments.
Maybe to confirm the in vivo utility the authors should use gastric or duodenal secretion from experimental animals of to evaluate the efficacy of this preparation with in vivo experiments.
Author Response
We appreciate the valuable advices that have helped us to improve the manuscript. Changes to the manuscript have been listed as follows.
1. Line 26 change has for have
“has” was changed to “have” in line 26.
2. Line 45 to thier properties
“thier” was changed to “their” in line 45.
3. Line 53 including exerting
“including exerting” was changed to “including” in line 53.
4. Line 140. In order to explanation
“explanation” was changed to “explain" in line 140.
5. Line 274 to prepared
“prepared” was changed to “prepare” in line 274.
6. Introduction I would avoid the term in gastrointestinal fluid since the study was done in vitro just changing the pH of the medium. Gastrointestinal fluid has a complex composition that cannot be compared with the vehicle used in these experiments.
“gastrointestinal fluid” has been changed to “phosphate buffer solution (pH 1.2 or 7.4)”.
Reviewer 3 Report
The manuscript entitled “Nano-in-micro delivery system prepared by co-axial air flow for oral delivery of conjugated linoleic acid” by Li, Q. et al describes the preparation of NiMDS with co axial flow method and their delivery. Please see comments below:
Does the authors see any difference by delivering Cs.NPs onto the gastrointestinal system without the reaction with sodium alginate treatment. Please include the results in the manuscript.
When the authors record the proton nmr of the chitosan and chitosan stearic acid in fig 2i, please include the CH2 protons integration with respect to chitosan which will also differentiates the degree of substitution in addition to elemental analysis results. Include revised figure for 2i is needed.
In fig 2j, the FTIR doesn’t show much difference between alginate versus NiMDS, please provide further evidence for encapsulation of particles.
As the polymer is amphiphilic, please provide experimental evidence for hydrophobic interactions in aqueous media.
In page 4, line 108, please provide evidence for hydrogen bonding interaction with nmr titrations.
Section 2.4 values are not matching with table 2, please correct and explain properly. Label the table2 properly.
For in vitro release study (fig 3a), It seems that the authors used pH 1.2 for only 3h, please provide the data until 10 h at pH 1.2 to see the difference with respect to pH 7.4 for 7 h? Any in vivo studies for its stability and release in the gastrointestinal environment.
The manuscript is written well with some typos and missing spaces. Please correct them.
Author Response
We appreciate the valuable advices that have helped us to improve the manuscript. Changes to the manuscript have been listed as follows.
1. Does the authors see any difference by delivering Cs.NPs onto the gastrointestinal system without the reaction with sodium alginate treatment. Please include the results in the manuscript.
The last sentence in conclusion was changed to “The chitosan nanoparticles embedded in alginate microparticle matrix showed less fragments, bigger size, higher mechanical strength, lower burst release and controlled-release in phosphate buffer solution (pH 1.2 or 7.4) than chitosan nanoparticles or alginate microparticles alone.”
2. When the authors record the proton nmr of the chitosan and chitosan stearic acid in fig 2i, please include the CH2 protons integration with respect to chitosan which will also differentiates the degree of substitution in addition to elemental analysis results. Include revised figure for 2i is needed.
The degree of substitution was different when using CH2 protons integration with respect to 3.0 ppm (H2) or 3.4~3.8 ppm (H3, H4, H6, H6,) respectively. So, we chose elemental analysis as the test method.
3. In fig 2j, the FTIR doesn’t show much difference between alginate versus NiMDS, please provide further evidence for encapsulation of particles.
The fluorescence micrograph of the NiMDS provide further evidence for encapsulation of nanoparticles.
In order to expressed clearly, “Figure 2b showed that the chitosan nanoparticles were uniformly distributed in microparticles” was changed to “Figure 2b showed that green fluorescence of FITC-chitosan nanoparticles uniformly dispersed in alginate microparticles, indicating successful encapsulate of nanoparticles into microparticles.” “fluorescence micrograph of the NiMDS” was changed to “fluorescence micrograph of the FITC labeled chitosan nanoparticles in alginate microparticles” in Figure 2.
4. As the polymer is amphiphilic, please provide experimental evidence for hydrophobic interactions in aqueous media.
“The critical aggregation concentration is defined as the threshold concentration of self-aggregation of amphiphilic polymers by intra- and/or intermolecular hydrophobic interactions. As shown in table 1, critical aggregation concentration decreased from 0.037 mg/ml to 0.027 mg/ml with the increasing of degree of substitution. This indicated that higher degree of substitution enhanced hydrophobic interactions and facilitated the formation of nanoparticles.” was added.
5. In page 4, line 108, please provide evidence for hydrogen bonding interaction with nmr titrations.
The hydrogen bonding was evidenced by FTIR measurement, and we are sorry that the NMR titration experiment can’t been finished at a short given time.
6. Section 2.4 values are not matching with table 2, please correct and explain properly. Label the table2 properly.
“EE”was changed to “loading efficiency” in section 2.4.
The data in the text and in table 2 has been matched.
7. For in vitro release study (fig 3a), It seems that the authors used pH 1.2 for only 3h, please provide the data until 10 h at pH 1.2 to see the difference with respect to pH 7.4 for 7 h? Any in vivo studies for its stability and release in the gastrointestinal environment.
The residence time of food is about 3h in the stomach, so we chose 3h in the experiment.
We preliminarily discussed stability and release properties in the phosphate buffer solution (pH 1.2 and 7.4). The in vivo studies have not been carried out yet, and we are appreciate your precious advice.
8. The manuscript is written well with some typos and missing spaces. Please correct them.
The missing spaces and grammatical errors have been corrected.
Round 2
Reviewer 3 Report
Please see below as highlighted in blue:
2. When the authors record the proton nmr of the chitosan and chitosan stearic acid in fig 2i, please include the CH2 protons integration with respect to chitosan which will also differentiates the degree of substitution in addition to elemental analysis results. Include revised figure for 2i is needed.
The degree of substitution was different when using CH2 protons integration with respect to 3.0 ppm (H2) or 3.4~3.8 ppm (H3, H4, H6, H6,) respectively. So, we chose elemental analysis as the test method.
It is weird that the author's proton nmr analysis is not matching with the elemental analysis data.
As the revised figure 2i, please show stearic acid proton nmr only at the bottom and chitosan only proton nmr as stacked nmr in the middle and then chitosan-stearic acid proton nmr on the top with appropriate integrations with respect to atleast stearic acid in the all the spectra.
3. In fig 2j, the FTIR doesn’t show much difference between alginate versus NiMDS, please provide further evidence for encapsulation of particles.
The fluorescence micrograph of the NiMDS provide further evidence for encapsulation of nanoparticles.
In order to expressed clearly, “Figure 2b showed that the chitosan nanoparticles were uniformly distributed in microparticles” was changed to “Figure 2b showed that green fluorescence of FITC-chitosan nanoparticles uniformly dispersed in alginate microparticles, indicating successful encapsulate of nanoparticles into microparticles.” “fluorescence micrograph of the NiMDS” was changed to “fluorescence micrograph of the FITC labeled chitosan nanoparticles in alginate microparticles” in Figure 2.
Fig 2b needs to be revised. Please show fluorescence micrograph of FITC labeled chitosan nanoparticles as bottom and show the fluorescence micrograph of FITC labled chitosan nanoparticles with alginate microparticles. Please insert the revised figure in the updated version.
Author Response
Thank you very much for attentively review, we have made corresponding modifications.
2. When the authors record the proton nmr of the chitosan and chitosan stearic acid in fig 2i, please include the CH2 protons integration with respect to chitosan which will also differentiates the degree of substitution in addition to elemental analysis results. Include revised figure for 2i is needed.
The degree of substitution was different when using CH2 protons integration with respect to 3.0 ppm (H2) or 3.4~3.8 ppm (H3, H4, H6, H6,) respectively. So, we chose elemental analysis as the test method.
It is weird that the author's proton nmr analysis is not matching with the elemental analysis data.
As the revised figure 2i, please show stearic acid proton nmr only at the bottom and chitosan only proton nmr as stacked nmr in the middle and then chitosan-stearic acid proton nmr on the top with appropriate integrations with respect to at least stearic acid in the all the spectra.
The NMR figures of chitosan-stearic acid, chitosan, stearic acid have been arranged from top to bottom.
And “FITC labeled chitosan nanoparticles (inserted picture)” has been added in figure 2.
3. In fig 2j, the FTIR doesn’t show much difference between alginate versus NiMDS, please provide further evidence for encapsulation of particles.
The fluorescence micrograph of the NiMDS provide further evidence for encapsulation of nanoparticles.
In order to expressed clearly, “Figure 2b showed that the chitosan nanoparticles were uniformly distributed in microparticles” was changed to “Figure 2b showed that green fluorescence of FITC-chitosan nanoparticles uniformly dispersed in alginate microparticles, indicating successful encapsulate of nanoparticles into microparticles.” “fluorescence micrograph of the NiMDS” was changed to “fluorescence micrograph of the FITC labeled chitosan nanoparticles in alginate microparticles” in Figure 2.
Fig 2b needs to be revised. Please show fluorescence micrograph of FITC labeled chitosan nanoparticles as bottom and show the fluorescence micrograph of FITC labled chitosan nanoparticles with alginate microparticles. Please insert the revised figure in the updated version.
The fluorescence micrograph of FITC labeled chitosan nanoparticles has been inserted.
And “chitosan and stearic acid” has been added in figure 2.